# Secretome Analysis of *Thermothelomyces thermophilus* LMBC 162 Cultivated with *Tamarindus indica* Seeds Reveals CAZymes for Degradation of Lignocellulosic Biomass

**DOI:** 10.3390/jof10020121

**Published:** 2024-02-01

**Authors:** Alex Graça Contato, Tiago Cabral Borelli, Marcos Silveira Buckeridge, Janet Rogers, Steven Hartson, Rolf Alexander Prade, Maria de Lourdes Teixeira de Moraes Polizeli

**Affiliations:** 1Departamento de Bioquímica e Imunologia, Faculdade de Medicina de Ribeirão Preto, Universidade de São Paulo, Ribeirão Preto 14049-900, SP, Brazil; alexgraca.contato@gmail.com; 2Department of Microbiology and Molecular Genetics, Oklahoma State University, Stillwater, OK 74078, USA; rolf.prade@okstate.edu; 3Departamento de Biologia Celular e Molecular e Bioagentes Patogênicos, Faculdade de Medicina de Ribeirão Preto, Universidade de São Paulo, Ribeirão Preto 14049-901, SP, Brazil; tiago.borelli@usp.br; 4Departamento de Botânica, Instituto de Biociências, Universidade de São Paulo, São Paulo 05508-090, SP, Brazil; msbuck@usp.br; 5Department of Biochemistry and Molecular Biology, Oklahoma State University, Stillwater, OK 74078, USA; janet.rogers@okstate.edu (J.R.); hartson.steve@gmail.com (S.H.); 6Departamento de Biologia, Faculdade de Filosofia, Ciências e Letras de Ribeirão Preto, Universidade de São Paulo, Ribeirão Preto 14040-901, SP, Brazil

**Keywords:** CAZymes, lignocellulosic biomass, secretome, tamarind seeds, *Thermothelomyces thermophilus*

## Abstract

The analysis of the secretome allows us to identify the proteins, especially carbohydrate-active enzymes (CAZymes), secreted by different microorganisms cultivated under different conditions. The CAZymes are divided into five classes containing different protein families. *Thermothelomyces thermophilus* is a thermophilic ascomycete, a source of many glycoside hydrolases and oxidative enzymes that aid in the breakdown of lignocellulosic materials. The secretome analysis of *T. thermophilus* LMBC 162 cultivated with submerged fermentation using tamarind seeds as a carbon source revealed 79 proteins distributed between the five diverse classes of CAZymes: 5.55% auxiliary activity (AAs); 2.58% carbohydrate esterases (CEs); 20.58% polysaccharide lyases (PLs); and 71.29% glycoside hydrolases (GHs). In the identified GH families, 54.97% are cellulolytic, 16.27% are hemicellulolytic, and 0.05 are classified as other. Furthermore, 48.74% of CAZymes have carbohydrate-binding modules (CBMs). Observing the relative abundance, it is possible to state that only thirteen proteins comprise 92.19% of the identified proteins secreted and are probably the main proteins responsible for the efficient degradation of the bulk of the biomass: cellulose, hemicellulose, and pectin.

## 1. Introduction

Plant biomass consists of proteins, lignin, holocellulose (a fraction composed of cellulose fibers wrapped in hemicellulose-pectin), ash, salts, and minerals [1]. The increase in agro-industrial activity has led to the accumulation of many lignocellulosic residues, such as wood and various agro-industrial residues around the world [2,3,4,5]. The economic interest in these residues has increased significantly in recent years since they are renewable and cheap, having the potential to produce and generate chemicals and bioenergy [6,7,8]. The conversion of lignocellulosic biomass into ethanol and other chemical compounds can be performed using a multi-enzyme system acting in synergism [9,10], and it is of fundamental importance to study different microorganisms and understand the secretion of the enzymes of interest that can be applied to these processes [11].

The analysis of the fungal secretome has gained great visibility since, through these studies, it is possible to know the proteins secreted by different microorganisms, especially carbohydrate-active enzymes (CAZymes), grown under different conditions [12,13,14,15]. Based on their protein sequence similarities and three-dimensional folding structure, CAZymes are classified into several hundred different enzyme protein families [16]. These enzymes are involved in many biological processes, and they are responsible for the degradation, synthesis, and modification of carbohydrates [17,18].

Thermophilic fungi are a promising source of new enzymes for cost-effective industrial applications, including abundant thermostable enzymes for biomass degradation and generation of chemicals and biofuels [19,20,21]. Among them, a fungus that is described in the literature as a source of many CAZymes, especially glycoside hydrolases (GHs) and oxidative enzymes that aid in the breakdown of lignocellulosic materials, is the thermophilic ascomycete *Thermothelomyces thermophilus* (formerly *Myceliophthora thermophila*) [22,23,24,25]. This filamentous fungus has been shown to be safe for large-scale production processes and can utilize cost-effective sources of plant biomass [26], as waste from the fruit pulp industry, especially tamarind seeds [27].

Tamarind (*Tamarindus indica* L.) is a fruit plant native to equatorial Africa, India, and Southeast Asia and grows in tropical and subtropical regions, with an ideal average temperature of 25 °C [28]. It consists of pulp and seeds with a hard coating. Seeds constitute 30–40% of the fruit, with a large proportion being an agricultural by-product [29]. According to Gonçalves et al. [30], the tamarind seed composition is 1.82 ± 0.01% ash, 33.07 ± 1.40% lignin, 33.31 ± 3.56% cellulose, and 10.45 ± 1.45% hemicellulose, proving to be a promising source for the detection of CAZymes.

Due to their constitution, tamarind seeds have been utilized for the cultivation of microorganisms to produce microbial enzymes that cleave lignocellulosic biomass or as substrates in assays to test enzymatic activity [31,32]. In addition, these seeds are rich in xyloglucan, which corresponds to about 40% of their dry mass [33]. In this context, this study aimed to report the elucidation of the secretome profile and categorization of CAZymes by function and family of the filamentous fungus *T. thermophilus* LMBC 162 cultivated by submerged fermentation using tamarind seeds as a carbon source, which is a residue from the fruit pulp industry [28]. Obtaining these data, it was determined by relative abundance which are the main proteins responsible for the degradation of the biomass bulk: cellulose, hemicellulose, and pectin.

## 2. Methods and Materials

### 2.1. Maintenance of the Fungus and Culture Medium

The fungus *T. thermophilus* LMBC 162 used in this work was isolated in Ribeirão Preto, SP, Brazil. Its identification and deposit in GenBank with the accession code MK559967.1 was described by Contato et al. [31]. The maintenance of the thermophilic microorganism was carried out through the inoculation of its spores in potato dextrose agar medium (PDA) (Sigma-Aldrich, Saint Louis, MO, USA), keeping it through successive transfers in glass tubes containing the same medium and incubating at the temperature of 40 °C. Afterward, the tubes were kept under refrigeration for up to 30 days.

### 2.2. Submerged Cultivation of T. thermophilus LMBC 162 for Protein Secretion Induction

The submerged cultivation was performed according to Contato et al. [30]. A solution with 10^6^–10^7^ spores/mL was prepared. The fungus was grown in test tubes and suspended in sterile distilled water, and its spores were counted in a microscope through a Neubauer chamber. The suspension was inoculated into 125 mL Erlenmeyer flasks with 25 mL of Khanna medium (Khanna’s salt solution [20×]: NH_4_NO_3_ (52.72%), KH_2_PO_4_ (34.27%), MgSO_4_·7H_2_O (9.54%), KCl (2.58%), MnSO_4_·H_2_O (0.36%), ZnSO_4_·H2O (0.18%), Fe_2_(SO_4_)_3_.6H_2_O (0.17%), CuSO_4_.5H_2_O (0.16%), distilled water q.s. (100 mL) (5.0 mL); yeast extract (0.1 g); carbon source (1.0 g); distilled water q.s. 100 mL) [34]. The medium was supplemented with 1% (*w*/*v*) of tamarind (*Tamarindus indica*, Fabaceae) seeds, which were previously pretreated (boiled in water, dried, and ground to 20 mesh) to secure the sanitary quality of the seeds and avoid the growth of other associated fungi. The Erlenmeyer flasks were incubated at 40 °C under static conditions for 72 h, the best conditions for protein induction described by Contato et al. [31] who showed that these conditions were substantially better than in shaken cultivation and with shorter (24 and 48 h) or longer (96 h) times for evaluating enzyme production. The cell biomass was filtered with the aid of a vacuum pump, and the filtrates were used as enzymatic extracts. This was performed in triplicate.

### 2.3. Protein Quantification

The proteins obtained in crude extract after cultivation were quantified using Bradford method [35]. Reactions were added with 160 µL of Bradford’s reagent and 40 µL of the enzymatic extracts and incubated for 5 min at room temperature. The absorbance was measured on a spectrophotometer (Shimadzu, Kyoto, Japan) at a wavelength of 595 nm, using bovine albumin as standard. The results were expressed in μg of protein/mL.

### 2.4. Sample Processing

The supernatant of *T. thermophilus* cultivated in tamarind seeds under submerged cultivation was collected by filtration after 72 h, concentrated by ultra-filtration (10,000 MWCO, PES membrane, Vivaspin, Littleton, CO, USA), rinsed twice with 5 mL of sodium acetate buffer 50 mM pH 5.0, and the proteins were separated using SDS-PAGE electrophoresis [36].

### 2.5. Characterization of the T. thermophilus LMBC 162 Using Liquid Chromatography-Tandem Mass Spectrometry (LC-MS/MS)

For secretome peptide mapping, concentrated *T. thermophilus* LMBC 162 was analyzed by reducing SDS-PAGE on 12% separation gels. For secretome LC–MS/MS analysis, 15–20 μg of identified secretome proteins were loaded onto an SDS-PAGE gel. Preparate PAGE gel electrophoresis was used to separate the protein secretomes from the complex carbohydrate and phenolic species accumulated in the supernatant. Proteins were briefly electrophoresed into the PAGE separating gel, with the electrophoresis being terminated after the bromophenol blue tracking dye had migrated 2–3 cm into the separating gel, stained with Coomassie blue, and the entire protein banding profile excised, processed for LC–MS/MS [37]. Isolated gel bands were reduced with 10 mM Tris (2-carboxyethyl) phosphine for 1 h at room temperature, alkylated by 10 mM using 2-iodoacetamide for 1 h at room temperature in the dark, and digested overnight at 37 °C with 8 μg/mL trypsin (Promega V5072, Madison, WI, USA) (relation proteins in bands: trypsin of 10 µg:mL trypsin/LysC) using 25 mM ammonium bicarbonate buffer, pH 8.0. Peptides were extracted from the gel segments with three sequential extractions at room temperature using 0.3 mL, 0.2 mL, and 0.2 mL of 0.5% trifluoracetic acid, respectively. After intermittent mixing at the vortex, they were dried in SpeedVac (ThermoFisher Scientific, Waltham, MA, USA). Finally, samples were desalted by solid phase extraction using C18 pipet tips, following the manufacturer’s recommendations (Agilent P/N A57003100, Agilent Technologies, Santa Clara, CA, USA). The desalted peptides were redissolved in 0.1% aqueous formic acid and injected onto a 75-micron × 50 cm capillary HPLC column packed with 2-micron C18 particles (Thermo P/N 164942, ThermoFisher Scientific, Waltham, MA, USA). Peptides were separated using a 60 min gradient of formic acid/acetonitrile with a flow rate of 250 nL/min and ionized in a Nanospray Flex (ThermoFisher Scientific, Waltham, MA, USA) ion source using stainless-steel emitters connected to a quadrupole-Orbitrap mass spectrometer (Fusion model, ThermoFisher Scientific, Waltham, MA, USA). Peptide ions were analyzed using a “high-low” “top-speed” data-dependent MS/MS strategy, wherein peptide precursors were analyzed at high resolution in the Orbitrap sector, selected for MS/MS using the quadrupole sector, fragmented by HCD in the ion routing multipole, followed by analysis of the fragment ions in the ion trap sector. MS/MS parameters used in the experiments are ions spray voltage (1900 W); capillary temperature (300 °C); mass range in full MS mode (375–1575 *m*/*z*); resolution setting for full mass MS scan, AGC target value, maximum injection time (120,000 nominal resolution, 4 × 10^5^ ions, 50 ms, respectively); number of peptides selected to be fragmented in each duty cycle (data-dependent acquisition limited only by cyclic rate, set at 5 s); value of normalized collision energy (32% HCD energy); resolution settings for MS/MS acquisition, AGC target value, maximum injection time (MS/MS analysis in the ion trap sector using a rapid scan rate, 5 × 10^4^ ions, dynamic injection timing limits wherein the system maximizes the injection times available for relative to the stated cycle time and maximizing sensitivity, respectively); charges of precursor ions excluded (below +2 or above +6 were excluded); and dynamic exclusion time (dynamic exclusion was set to 45 s).

Each sample was analyzed twice by LC-MS/MS, and the two RAW data files were specified as a single sample for database searching using MaxQuant (version v2.0.1.0, Max-Planck-Institute of Biochemistry, Planegg, Germany) [38]. Spectra were searched against a database of 18,464 *T. thermophilus* protein sequences downloaded from NCBI on 27 May 2022, using *Thermothelomyces* as a genus search term. Searches were annotated using Python version v3.11 (Python Software Foundation, Wilmington, DE, USA) to annotate NCBI *T. thermophilus* ID’s by transferring annotations from related curated proteins at Uniprot (https://www.uniprot.org/ accessed on 14 December 2022). Sequences with a false discovery rate (FDR or q-value) greater than 0.00 were removed from the analysis. Finally, we identified conserved CAZy domains using Hidden Markov Models (HMM) profiles available on the dbCAN2 web platform (https://bcb.unl.edu/dbCAN2/index.php accessed on 14 December 2022). Only domains with e-values > 10^−17^ and coverage > 0.35 were considered.

## 3. Results and Discussion

### Analysis of Secretome Protein Composition

To characterize the secretome of *T. thermophilus* LMBC 162, the supernatants of cultures were collected and analyzed using LC-MS/MS searching against a database of *Thermothelomyces* sequences downloaded from the NCBI. The identified proteins were annotated by searching the *T. thermophilus* sequences against curated protein sequences available in the Uniprot/Swiss-Prot database. Our analysis identified 79 proteins in the *T. thermophilus* LMBC 162 secretome (all non-anchored extracellular proteins). Taking into account the quantification of these proteins through their relative abundance referenced by the IBAQ value (sum of all the peptides intensities divided by the number of observable peptides of a protein), we found five diverse classes of CAZymes: 5.55% auxiliary activity (AAs); 2.58% carbohydrate esterases (CEs); 20.58% polysaccharide lyases (PLs); and 71.29% glycoside hydrolases (GHs), which were 54.97% cellulolytic GHs, 16.27% hemicellulolytic GHs, and 0.05 classified as other GHs. Furthermore, 48.74% of CAZymes have carbohydrate-binding modules (CBMs) (Figure 1). These values are consistent with others shown in the literature for other filamentous fungi [24,39].

***Auxiliary activity (AA) enzymes.*** The AAs are families of catalytic proteins that are potentially involved in plant cell degradation through an ability to help the original glycoside hydrolase, polysaccharide lyase, and carbohydrate esterase enzymes to gain access to the carbohydrates comprising the plant cell wall [40]. Among the 17 auxiliary activity enzymes, we observed eight (8) lytic polysaccharide monooxygenases (LPMOs), with the majority being from the AA9 CAZy domain (Table 1). Analyzing quantitatively by the relative abundance (IBAQ value), we identified that the AAs correspond to 5.55% of the proteins detected in the secretome analysis. Other AA CAZy domains found are AA3, AA5, AA7, AA8, AA12, and AA13. These results corroborate studies that, when performing the secretome profile of this microorganism with other cultivated sources, also verified the presence of these oxidative enzymes [22,24].

***Carbohydrate esterases (CEs).*** CE catalyzes the de-O- or de-N-acylation by removing the ester decorations from carbohydrates. They represent biocatalysts important for bioconversion of plant biomass and saccharification of plant cell wall polysaccharide fractions that have not gone through an alkaline pretreatment or process that would destroy the ester linkages [41]. In this study, seven (7) CEs were found, corresponding to 2.58% of the proteins of the *T. thermophilus* LMBC 162 secretome cultivated using submerged fermentation with tamarind seeds (Table 2). These values are consistent with those shown by Rocha et al. [42], who found a relative abundance of 2% CEs in the *Trichoderma harzianum* secretome when cultivated on sugarcane bagasse, and with the study by Machado et al. [39] who obtained 3.4% of esterases in the *Trametes versicolor* secretome cultivated on microcrystalline cellulose. Among the CEs identified in this work, two were highlighted with 2.29% relative abundance, that is, 88.75% of the CEs. They are an acetylesterase CE16, an enzyme that catalyzes the conversion of acetate esters and water into alcohols and acetate [43], and a pectinesterase CE8 (accession number G2QLD0) that catalyzes the de-esterification of pectin into pectate and methanol [44]. They correspond to 1.19% and 1.10% of relative abundance, respectively. Other CE CAZy domains found are CE1, CE3, CE5, and CE12.

***Polysaccharide lyases (PLs).*** PLs are a group of enzymes that cleave uronic acid-containing polysaccharide chains via a β-elimination mechanism to generate an unsaturated hexenuronic acid residue and a new reducing end [45]. In this work, six (6) PLs were visualized, which correspond to 20.58% of the secretome. However, there is a huge emphasis on a particular protein from the PL1 family, accession number (G2QH79) in the UniProt/Swiss-Prot database and hypothetical molecular weight of (34 kDa) (Table 3). This PL alone corresponds to 19.95% of relative abundance, which is 96.94% of all identified PLs. These values are higher than the 9% seen by Verma et al. [46] in the secretome of the fungal phytopathogen *Ascochyta rabiei* and the 7% of those seen by Rubio et al. [47] for *Aspergillus nidulans*. However, this is consistent with the study by dos Santos et al. [24] who used another strain of *Myceliophthora thermophila* and cultivated on lignocellulosic residues. Other PL CAZy domains found are PL3 and PL4. The classes of CEs and PLs are mainly responsible for the degradation of pectin, one of the main components of the cell wall of plants [48].

***Cellulolytic glycoside hydrolases (GHs).*** The GHs are enzymes that catalyze the hydrolysis of the glycosidic linkage of glycosides, leading to the formation of a sugar hemiacetal or hemiketal and the corresponding free aglycon [49]. The GHs that cleave sugars from cellulose are named cellulolytic glycoside hydrolases, and those that cleave sugars from hemicellulose are named hemicellulolytic glycoside hydrolases. Of the CAZymes found in the secretome of *T. thermophilus* LMBC 162, forty-nine (49) are GHs, corresponding to 71.29% in relative abundance. Among them, the majority are those that breakdown cellulose. They account for 54.97% of the CAZymes produced in the secretome (Table 4). One cellobiohydrolase GH7 and a glucoside hydrolase from the GH7 CAZy domain are the main proteins found. They can be seen with accession numbers G2Q665 and G2QNN8 in the UniProt/Swiss-Prot database. The hypothetical molecular weight of each is 56 kDa and 49 kDa, respectively. Other GHs CAZy domains found are GH6, GH15, GH31, and GH45. The value of cellulolytic GHs found is like those seen by Machado et al. [39] who found 48.1% for *Phanerochaete chrysosporium* and 48.0% for *T. versicolor*.

***Hemicellulolytic glycoside hydrolases (GHs).*** Regarding the hemicellulolytic GHs found (Table 5), they correspond to 16.27% in relative abundance, once again being equivalent to values reported by others [24,39]. The hemicellulolytic GHs belong to the CAZy domains: GH2, GH3, GH10, GH11, GH16, GH26, GH27, GH43, GH47, GH55, GH62, GH74, GH76, GH79, GH92, GH93, GH125, GH131, and GH135. The hemicellulolytic GHs with the highest relative abundance are a xylanase GH10 with a hypothetical molecular weight of (45 kDa) and accession number (G2QJ91) in the Uniprot/Swiss-Prot database, with 2.11% of relative abundance; a xylanase GH11 with a hypothetical molecular weight of (24 kDa) and accession number (G2Q4M3), with a relative abundance of 2.59%; a xyloglucanase GH74 of hypothetical molecular weight of (79 kDa) and accession number (G2QHR7), with 5.39% of relative abundance; and an *exo*-α-L-1,5-arabinanase GH93 with a hypothetical molecular weight of (42 kDa) and accession number (G2Q5Q6), with a relative abundance of 2.30%. The GH74 found in this work showed a hypothetical molecular weight corresponding to the xyloglucanase found by Berezina et al. [50], who expressed a GH74 xyloglucanase from *M. thermophila* in *Pichia pastoris*. Another relevant factor is that this secretome was performed in tamarind seeds, which are rich in xyloglucan [32], thus proving why the GH74 was produced with the highest relative abundance. Regarding the GH93 family, it is known to hydrolyze linear α-1,5-L-arabinan [51].

In the analysis of *T. thermophilus* LMBC 162 secreted proteins belonging to the hemicellulolytic GH family, two uncharacterized proteins were determined. The GH131 family are β-glucanases that exhibit activity for a wide range of β-glucan polysaccharides, including laminarin, curdlan, lichenan, and cellulosic derivatives [52], while the GH135 family has disclosed fungal glycoside hydrolases with the ability to degrade the fungal heteropolysaccharide galactosaminogalactan [53].

Other relevant GHs found in *T. thermophilus* LMBC 162 secretome are the GH3, GH10, GH16, GH43, and GH55 CAZy domains. GH3 removes single glycoside residues from the non-reducing ends of their substrates and currently groups together *exo*-acting β-D-glucosidases, α-L-arabinofuranosidases, β-D-xylopyranosidases, N-acetyl-β-D-glucosaminidases, and N-acetyl-β-D-glucosaminide phosphorylases [54]; GH10 are *endo*-β-1,4-xylanases [55]; GH16 are active on β-1,4 or β-1,3 glycosidic bonds in various glucans and galactans, which include lichenases (EC 3.2.1.73) [56]. GH43 are α-L-arabinofuranosidases, endo-α-L-arabinanases (or *endo*-processive arabinanases), and β-D-xylosidases [57]; and GH55 consists exclusively of β-1,3-glucanases, including both *exo*-and *endo*-enzymes [58].

***Other glycoside hydrolases (GHs).*** Table 6 shows GHs identified that breakdown other components. Only two GHs were detected: a trehalase GH37, which corresponds to 0.01% of the secretome, and an α-L-fucosidase GH95, corresponding to 0.04% of the secretome. The enzymes from family GH37 have been shown to hydrolyze the α-1,1 bound trehalose (α-D-glucopyranosyl-(1→1)-α-D-glucopyranoside) into two molecules of D-glucose [59], while the α-L-fucosidase GH95 hydrolyzes α-Fuc-1,2-Gal linkages attached at the non-reducing ends of oligosaccharides [60].

***Carbohydrate binding modules (CBMs).*** CBMs are protein domains found in CAZymes, whose main role is to recognize and bind specifically to carbohydrates. The consequences of this event result in different functions, such as increased hydrolysis of insoluble substrates, bringing the catalytic domain closer to the substrate, polysaccharide structure disruption, and cell surface protein anchoring [61]. In the secretome of *T. thermophilus* LMBC 162, sixteen (16) CAZymes have CBMs (Table 7). In sum, CBMs are present in 48.74% of the proteins found in the *T. thermophilus* secretome profile, a value like those shown in the literature for other microorganisms [39].

Observing the iBAQ/total iBAQ values as a percentage, it is possible to state that only thirteen (13) proteins comprise 92.19% of the identified proteins secreted, and they are probably the main proteins responsible for the degradation of the bulk of the biomass: cellulose, hemicellulose, and pectin (Table 8). The most abundant protein in the secretome of *T. thermophilus* LMBC 162 (31.43%) was a cellobiohydrolase, like the secretome of *Trichoderma reesei* RUT C30, where a cellobiohydrolase is the most abundant protein [12]. However, the presence of other enzymes, such as β-xylanase, lytic polysaccharide monooxygenase, and pectinesterase, was reported. Nevertheless, one of the limitations of shotgun proteomics is incomplete sequence coverage when using only one protease. Therefore, there is a possibility that other proteins, such as small proteins due to the few theoretical peptides produced in digestion, were not detected [62].

## 4. Conclusions

The secretome analysis of *T. thermophilus* LMBC 162 cultivated by submerged fermentation with tamarind seeds, an abundant residue from the fruit pulp industry, reveals seventy-nine (79) CAZymes diversified into the five classes of CAZy database: 5.55% AAs; 1.48% CBMs; 2.58% CEs, 20.58% PLs; and 70.55% GHs, which are 54.97% cellulolytic GHs, 15.51% hemicellulolytic GHs, and 0.05 classified as other GHs. Between them, sixteen (16) CAZymes have CBMs, protein domains found in CAZymes, whose main role is to recognize and bind specifically to carbohydrates. In sum, CBMs are present in 48.74% of the proteins found in the *T. thermophilus* secretome profile, a value like those shown in the literature for other microorganisms. Observing the relative abundance, it is possible to state that only thirteen (13) proteins comprise 92.19% of the identified proteins secreted, and they are probably the main proteins responsible for the degradation of the bulk of the biomass: cellulose, hemicellulose, and pectin. The findings of this work allow us to say that tamarind seeds are a residue option for the identification and production of lignocellulosic CAZymes.

## Figures and Tables

**Figure 1 jof-10-00121-f001:**
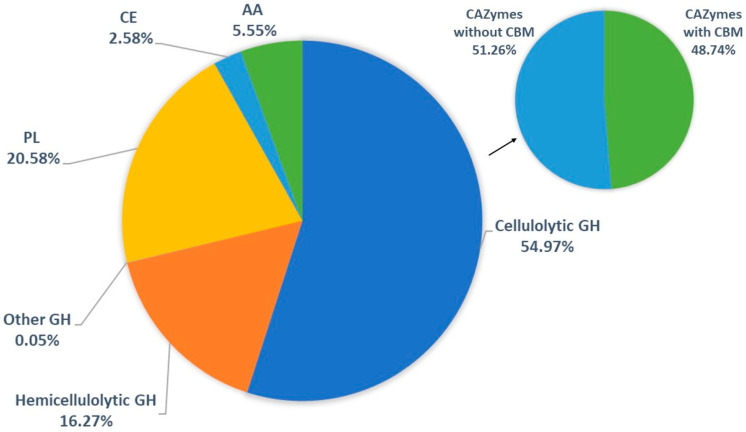
CAZymes from secretome analysis of *T. thermophilus* LMBC 162. Graph units are based on relative abundance according to data from the LC-MS/MS. Auxiliary activity (AA), carbohydrate-binding module (CBM), carbohydrate esterases (CEs), polysaccharide lyases (PLs), and glycoside hydrolases (GHs).

**Table 1 jof-10-00121-t001:** LC-MS/MS secretome analysis for auxiliary activity (AA) enzymes.

MS/MS View: Identified Proteins	Accession Number		Molecular Weight (kDa) ^a^	CAZy Domain	BLAST E-Value ^b^	iBAQ ^c^	iBAQ/Total iBAQ (%)
	UniProt/Swiss-Prot Database	NCBI Database					
glucose–methanol–choline GMC oxidoreductase	G2PZJ2	XP_003660923.1	69	AA3	9.5 × 10^−153^	11699000	0.35
cellobiose dehydrogenase	G2QGP4	XP_003663851.1	68	AA3—CBM1	2.2 × 10^−76^	231140	0.73
cellobiose dehydrogenase	G2QNS9	XP_003666548.1	57	AA3	7 × 10^−197^	210360	0.01
glyoxal oxidase	G2Q335	XP_003658743.1	105	AA5	4.8 × 10^−142^	86049	0.01
FAD linked oxidase	G2QG48	XP_003663758.1	54	AA7	5.9 × 10^−92^	818280	0.02
FAD linked oxidase	G2Q654	XP_003660778.1	69	AA7	1.8 × 10^−45^	98150	0.01
cellobiose dehydrogenase	G2QFY4	XP_003664543.1	84	AA8	6 × 10^−65^	984050	0.03
cellobiose dehydrogenase	A9XK88	XP_003663382.1	89	AA8—CBM1	2.5 × 10^−71^	846980	0.02
lytic polysaccharide monooxygenase G	G2Q4M0	XP_003659754.1	32	AA9—CBM1	1.7 × 10^−67^	65875000	1.93
lytic polysaccharide monooxygenase B	G2QCJ3	XP_003663414.1	32	AA9—CBM1	1.4 × 10^−74^	46761000	1.36
lytic polysaccharide monooxygenase H	G2Q9T3	XP_003661787.1	35	AA9—CBM1	4.3 × 10^−71^	25188000	0.74
lytic polysaccharide monooxygenase J	G2Q7A5	XP_003661261.1	26	AA9	1.4 × 10^−74^	6202800	0.18
lytic polysaccharide monooxygenase F	G2QK49	XP_003665200.1	24	AA9	2.7 × 10^−69^	3043500	0.09
lytic polysaccharide monooxygenase I	G2Q9F7	XP_003661661.1	31	AA9—CBM1	2 × 10^−71^	445310	0.01
lytic polysaccharide monooxygenase A	G2QI82	XP_003665516.1	24	AA9	3.8 × 10^−72^	153330	0.01
pyrroloquinoline quinone-dependent oxidoreductase	G2QES6	XP_003664200.1	54	AA12	1 × 10^−171^	542100	0.02
lytic polysaccharide monooxygenase	G2QH80	XP_003663985.1	20	AA13	2.5 × 10^−65^	1054800	0.03

^a^ Hypothetical molecular weight of the proteins, ^b^ BLAST E-value is the number of expected hits of similar quality (score) that could be found just by chance, ^c^ the iBAQ corresponds to the sum of all the peptide intensities divided by the number of observable peptides of a protein.

**Table 2 jof-10-00121-t002:** LC-MS/MS secretome analysis for carbohydrate esterases (CEs).

MS/MS View: Identified Proteins	Accession Number		Molecular Weight (kDa) ^a^	CAZy Domain	BLAST E-Value ^b^	iBAQ ^c^	iBAQ/Total iBAQ (%)
	UniProt/Swiss-Prot Database	NCBI Database					
acetylxylan esterase	G2QD29	XP_003663492.1	34	CE1	1.5 × 10^−23^	5647500	0.17
lipase	G2QGB0	XP_003664615.1	26	CE3	4.5 × 10^−54^	135180	0.01
acetylxylan esterase	G2QJ94	XP_003665705.1	32	CE5—CBM1	1.3 × 10^−41^	2033500	0.06
pectinesterase	G2QLD0	XP_003666007.1	36	CE8	1.1 × 10^−75^	40224000	1.19
pectinesterase	G2QMM2	XP_003666447.1	35	CE8	7 × 10^−79^	760070	0.02
rhamnogalacturonan acetylesterase	G2QMH3	XP_003666398.1	28	CE12	6.2 × 10^−45^	1178900	0.03
acetylesterase	G2QJ27	XP_003664847.1	32	CE16	6.1 × 10^−101^	37435000	1.10

^a^ Hypothetical molecular weight of the proteins, ^b^ BLAST E-value is the number of expected hits of similar quality (score) that could be found just by chance, ^c^ the iBAQ corresponds to the sum of all the peptide intensities divided by the number of observable peptides of a protein.

**Table 3 jof-10-00121-t003:** LC-MS/MS secretome analysis for polysaccharide lyases (PLs).

MS/MS View: Identified Proteins	Accession Number		Molecular Weight (kDa) ^a^	CAZy Domain	BLAST E-Value ^b^	iBAQ ^c^	iBAQ/Total iBAQ (%)
	UniProt/Swiss-Prot Database	NCBI Database					
pectate lyase	G2QH79	XP_003663984.1	34	PL1	8.3 × 10^−82^	676130000	19.95
pectate lyase	G2Q1K5	XP_003660241.1	35	PL1	1.3 × 10^−94^	10096000	0.30
pectate lyase	G2QMM3	XP_003666448.1	33	PL1	1.9 × 10^−45^	1454500	0.04
pectate lyase	G2QG50	XP_003663760.1	40	PL1	6.5 × 10^−86^	346420	0.01
pectate lyase	G2QG74	XP_003664579.1	26	PL3	2.3 × 10^−82^	2939800	0.09
rhamnogalacturonase B	G2QFG7	XP_003664441.1	58	PL4	4.4 × 10^−211^	6525300	0.19

^a^ Hypothetical molecular weight of the proteins, ^b^ BLAST E-value is the number of expected hits of similar quality (score) that could be found just by chance, ^c^ the iBAQ corresponds to the sum of all the peptide intensities divided by the number of observable peptides of a protein.

**Table 4 jof-10-00121-t004:** LC-MS/MS secretome analysis for glycoside hydrolases (GHs) that breakdown cellulose.

MS/MS View: Identified Proteins	Accession Number		Molecular Weight (kDa) ^a^	CAZy Domain	BLAST E-Value ^b^	iBAQ ^c^	iBAQ/Total iBAQ (%)
	UniProt/Swiss-Prot Database	NCBI Database					
cellobiohydrolase	G2QA39	XP_003661032.1	51	GH6—CBM1	8.2 × 10^−97^	167100000	4.93
cellobiohydrolase	G2QFW6	XP_003664525.1	42	GH6	2.7 × 10^−91^	5419800	0.16
cellobiohydrolase	G2Q665	XP_003660789.1	56	GH7—CBM1	5.5 × 10^−198^	1066100000	31.43
glycoside hydrolase	G2QNN8	XP_003666507.1	49	GH7	2 × 10^−175^	464770000	13.71
endoglucanase	G2QCS4	XP_003663441.1	49	GH7—CBM1	1 × 10^−138^	142270000	4.20
endoglucanase	G2QGA1	XP_003664606.1	49	GH7	3.2 × 10^−153^	9135400	0.27
glycoside hydrolase	G2Q359	XP_003658767.1	49	GH7	1.1 × 10^−190^	137980	0.01
glucan 1,4-α-glucosidase	G2QPS0	XP_003666828.1	67	GH15—CBM20	2.4 × 10^−74^	2191000	0.06
glycoside hydrolase	G2QAE3	XP_003661084.1	103	GH31	1.2 × 10^−148^	28072	0.01
endoglucanase	G2Q0Y0	XP_003659323.1	24	GH45	1.1 × 10^−92^	6393300	0.19

^a^ Hypothetical molecular weight of the proteins, ^b^ BLAST E-value is the number of expected hits of similar quality (score) that could be found just by chance, ^c^ the iBAQ corresponds to the sum of all the peptide intensities divided by the number of observable peptides of a protein.

**Table 5 jof-10-00121-t005:** LC-MS/MS secretome analysis for glycoside hydrolases (GHs) that breakdown hemicellulose.

MS/MS View: Identified Proteins	Accession Number		Molecular Weight (kDa) ^a^	CAZy Domain	BLAST E-Value ^b^	iBAQ ^c^	iBAQ/Total iBAQ (%)
	UniProt/Swiss-Prot Database	NCBI Database					
β-galactosidase	G2QGS8	XP_003664680.1	96	GH2	5.7 × 10^−120^	1017800	0.03
β-glucosidase	G2QDN2	XP_003663588.1	78	GH3	7.1 × 10^−61^	1628000	0.05
β-glucosidase	G2QCQ3	XP_003663420.1	95	GH3	2.8 × 10^−61^	880340	0.03
xylan 1,4-β-xylosidase	G2QKP9	XP_003665776.1	90	GH3	2.4 × 10^−59^	365920	0.01
xylanase	G2QJ91	XP_003665702.1	45	GH10—CBM1	1.1 × 10^−99^	71371000	2.11
xylanase	G2QGN6	XP_003663843.1	42	GH10	1.3 × 10^−91^	1989200	0.06
xylanase	G2QG07	XP_003664565.1	36	GH10	2.7 × 10^−106^	528610	0.02
xylanase	G2Q4M3	XP_003659757.1	24	GH11	2 × 10^−53^	87638000	2.59
glycoside hydrolase	G2QLD1	XP_003666008.1	31	GH16	2.1 × 10^−53^	8443400	0.25
glycoside hydrolase	G2QHP5	XP_003664150.1	41	GH16	2.5 × 10^−71^	2031600	0.06
glycoside hydrolase	G2Q2L1	XP_003658675.1	44	GH16	2.9 × 10^−71^	37364	0.01
mannan endo-1,4-β-mannosidase A	G2Q4H7	XP_003658915.1	53	GH26—CBM35	2.3 × 10^−35^	8956300	0.26
α-galactosidase	G2QNU8	XP_003667369.1	45	GH27	1.1 × 10^−56^	3528300	0.10
α-L-arabinofuranosidase 1	G2QFK1	XP_003663668.1	35	GH43	1.4 × 10^−97^	23454000	0.69
arabinanase	G2QCC8	XP_003662548.1	61	GH43	2 × 10^−132^	18453000	0.54
arabinan *endo*-1,5-α-L-arabinosidase	G2QFK0	XP_003663667.1	35	GH43	1.5 × 10^−125^	4569300	0.13
arabinanase	G2QDD9	XP_003663549.1	49	GH43—CBM35	1.9 × 10^−84^	4257400	0.13
arabinanase	G2QQ09	XP_003666917.1	55	GH43	1.9 × 10^−115^	2470100	0.07
arabinanase	G2QHQ9	XP_003664164.1	39	GH43	5.4 × 10^−100^	1494900	0.04
α-1,2-mannosidase	G2QHL4	XP_003664119.1	58	GH47	1.5 × 10^−136^	446310	0.01
*exo*-β-1,3-glucanase	G2QCT8	XP_003663454.1	84	GH55	0	2534300	0.07
*exo*-β-1,3-glucanase	G2PZK7	XP_003660938.1	95	GH55	1.6 × 10^−259^	1030200	0.03
*exo*-β-1,3-glucanase	G2QF48	XP_003664322.1	82	GH55	6.9 × 10^−292^	241200	0.01
*exo*-β-1,3-glucanase	G2QIM4	XP_003665591.1	82	GH55	4.7 × 10^−217^	27566	0.01
α-L-arabinofuranosidase	G2QJQ6	XP_003665058.1	35	GH62	2.1 × 10^−131^	6301300	0.19
α-L-arabinofuranosidase	G2QLV4	XP_003666179.1	40	GH62—CBM1	1.7 × 10^−125^	295260	0.01
1,3-β-glucanosyltransferase	G2QN92	XP_003667210.1	52	GH72	9.3 × 10^−122^	745680	0.02
1,3-β-glucanosyltransferase	G2QAD1	XP_003661926.1	57	GH72	1.8 × 10^−128^	603890	0.02
xyloglucanase	G2QHR7	XP_003664172.1	79	GH74	1.2 × 10^−22^	183060000	5.39
α-mannanase	G2QL30	XP_003665907.1	46	GH76	1.6 × 10^−96^	39672	0.01
glycoside hydrolase	G2QJT2	XP_003665083.1	51	GH79	7.1 × 10^−64^	1411000	0.04
α-mannosidase	G2Q1N1	XP_003660267.1	90	GH92	3.4 × 10^−160^	135570	0.01
*exo*-α-L-1,5-arabinanase	G2Q5Q6	XP_003660737.1	42	GH93	2.8 × 10^−108^	78089000	2.30
glycoside hydrolase	G2QGJ6	XP_003664651.1	55	GH125	3.4 × 10^−145^	106580	0.01
glycoside hydrolase	G2QNK3	XP_003667321.1	37	GH131—CBM1	1.2 × 10^−111^	25649000	0.76
uncharacterized protein MYCTH_2295704	G2Q5V8	XP_003659079.1	32	GH131	3.6 × 10^−76^	3439200	0.10
uncharacterized protein MYCTH_2301831	G2QAC0	XP_003661915.1	33	GH135	1.4 × 10^−43^	3426300	0.10

^a^ Hypothetical molecular weight of the proteins, ^b^ BLAST E-value is the number of expected hits of similar quality (score) that could be found just by chance, ^c^ the iBAQ corresponds to the sum of all the peptide intensities divided by the number of observable peptides of a protein.

**Table 6 jof-10-00121-t006:** LC-MS/MS secretome analysis for glycoside hydrolases (GHs) that breakdown other components.

MS/MS View: Identified Proteins	Accession Number		Molecular Weight (kDa) ^a^	CAZy Domain	BLAST E-Value ^b^	iBAQ ^c^	iBAQ/Total iBAQ (%)
	UniProt/Swiss-Prot Database	NCBI Database					
trehalase	G2PZS2	XP_003658392.1	78	GH37	1.4 × 10^−159^	154860	0.01
α-L-fucosidase	G2QDI5	XP_003662742.1	91	GH95	7.4 × 10^−209^	1873100	0.04

^a^ Hypothetical molecular weight of the proteins, ^b^ BLAST E-value is the number of expected hits of similar quality (score) that could be found just by chance, ^c^ the iBAQ corresponds to the sum of all the peptide intensities divided by the number of observable peptides of a protein.

**Table 7 jof-10-00121-t007:** LC-MS/MS secretome analysis for carbohydrate binding modules (CBMs).

MS/MS View: Identified Proteins	Accession Number		CAZy Domain	BLAST E-Value ^a^	iBAQ ^b^	iBAQ/Total iBAQ (%)
	UniProt/Swiss-Prot Database	NCBI Database				
cellobiose dehydrogenase	G2QGP4	XP_003663851.1	AA3—CBM1	2.2 × 10^−76^	231140	0.73
cellobiose dehydrogenase	A9XK88	XP_003663382.1	AA8—CBM1	2.5 × 10^−71^	846980	0.02
lytic polysaccharide monooxygenase G	G2Q4M0	XP_003659754.1	AA9—CBM1	1.7 × 10^−67^	65875000	1.93
lytic polysaccharide monooxygenase B	G2QCJ3	XP_003663414.1	AA9—CBM1	1.4 × 10^−74^	46761000	1.36
lytic polysaccharide monooxygenase H	G2Q9T3	XP_003661787.1	AA9—CBM1	4.3 × 10^−71^	25188000	0.74
lytic polysaccharide monooxygenase I	G2Q9F7	XP_003661661.1	AA9—CBM1	2 × 10^−71^	445310	0.01
acetylxylan esterase	G2QJ94	XP_003665705.1	CE5—CBM1	1.3 × 10^−41^	2033500	0.06
cellobiohydrolase	G2QA39	XP_003661032.1	GH6—CBM1	8.2 × 10^−97^	167100000	4.93
cellobiohydrolase	G2Q665	XP_003660789.1	GH7—CBM1	5.5 × 10^−198^	1066100000	31.43
endoglucanase	G2QCS4	XP_003663441.1	GH7—CBM1	1 × 10^−138^	142270000	4.20
glucan 1,4-α-glucosidase	G2QPS0	XP_003666828.1	GH15—CBM20	2.4 × 10^−74^	2191000	0.06
xylanase	G2QJ91	XP_003665702.1	GH10—CBM1	1.1 × 10^−99^	71371000	2.11
mannan endo-1,4-β-mannosidase A	G2Q4H7	XP_003658915.1	GH26—CBM35	2.3 × 10^−35^	8956300	0.26
arabinanase	G2QDD9	XP_003663549.1	GH43—CBM35	1.9 × 10^−84^	4257400	0.13
α-L-arabinofuranosidase	G2QLV4	XP_003666179.1	GH62—CBM1	1.7 × 10^−125^	295260	0.01
glycoside hydrolase	G2QNK3	XP_003667321.1	GH131—CBM1	1.2 × 10^−111^	25649000	0.76

^a^ BLAST E-value is the number of expected hits of similar quality (score) that could be found just by chance, ^b^ the iBAQ corresponds to the sum of all the peptide intensities divided by the number of observable peptides of a protein.

**Table 8 jof-10-00121-t008:** Proteins comprising 92.19% of the identified secreted proteins and classified according to their relative abundance (IBAQ/Total IBAQ) and with which part of the biomass they degrade: cellulose, hemicellulose, or pectin.

MS/MS View: Identified Proteins	Accession Number		Molecular Weight (kDa) ^a^	CAZy Domain	iBAQ/Total iBAQ (%) ^b^	Degraded Biomass
	UniProt/Swiss-Prot Database	NCBI Database				
cellobiohydrolase	G2Q665	XP_003660789.1	56	GH7—CBM1	31.43	cellulose
polysaccharide lyase	G2QH79	XP_003663984.1	34	PL1	19.95	pectin
glycoside hydrolase	G2QNN8	XP_003666507.1	49	GH7	13.71	cellulose
xyloglucanase	G2QHR7	XP_003664172.1	79	GH74	5.39	hemicellulose
cellobiohydrolase	G2QA39	XP_003661032.1	51	GH6—CBM1	4.93	cellulose
endoglucanase	G2QCS4	XP_003663441.1	49	GH7—CBM1	4.20	cellulose
xylanase	G2Q4M3	XP_003659757.1	24	GH11	2.59	hemicellulose
*exo*-α-L-1,5-arabinanase	G2Q5Q6	XP_003660737.1	42	GH93	2.30	hemicellulose
xylanase	G2QJ91	XP_003663843.1	45	GH10—CBM1	2.11	hemicellulose
lytic polysaccharide monooxygenase G	G2Q4M0	XP_003659754.1	32	AA9—CBM1	1.93	cellulose
lytic polysaccharide monooxygenase B	G2QCJ3	XP_003663414.1	32	AA9—CBM1	1.36	cellulose
pectinesterase	G2QLD0	XP_003666007.1	36	CE8	1.19	pectin
acetylesterase	G2QJ27	XP_003664847.1	32	CE16	1.10	pectin

^a^ Hypothetical molecular weight of the proteins, ^b^ the iBAQ corresponds to the sum of all the peptide intensities divided by the number of observable peptides of a protein.

## Data Availability

The datasets generated during and/or analyzed during the current study are available from the corresponding author on reasonable request.

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
