# Peer review of "Secretome Analysis of Thermothelomyces thermophilus LMBC 162 Cultivated with Tamarindus indica Seeds Reveals CAZymes for Degradation of Lignocellulosic Biomass"

_jof, 2024, doi:10.3390/jof10020121_

Round 1

Reviewer 1 Report

Comments and Suggestions for Authors

In this paper, carbohydrate-active enzymes (CAZymes) by Thermothelomyces thermophilus cultivated with Tamarindus indica seeds were investigated using the analysis of the secretome. Although some results seemed to be interesting, a few questions have to be clarified.

1. In Section 2.2, 0.1% yeast extract and 1% tamarind seeds were added into the medium. Because the addition of different nitrogen-containing substances in the medium can induce the production of different enzyme systems, and tamarind seeds also contain high protein content, these protein nutrients may affect the production of enzyme types. How to solve this problem?

2. As the title of the paper is for revealing CAZymes for Degradation of llignocellulosic biomass”, why not directly add tamarind carbohydrate to the chemical synthesis medium to avoid interference caused by the presence of protein in tamarind seeds?

3. In line 102-103, the Erlenmeyer flasks were incubated at 40 ℃ under static conditions for 72 h to analyze the composition of related enzymes, which was the best conditions for protein induction in the previous studies, and the data to prove this result should be reflected. 

Comments on the Quality of English Language

None

Author Response

January 06th, 2024

Dear Mr. Demir Li,

Assistant Editor, Journal of Fungi,

Find attached the revised version of the manuscript entitled “Secretome analysis of Thermothelomyces thermophilus LMBC 162 cultivated with Tamarindus indica seeds reveals CAZymes for degradation of lignocellulosic biomass”. We appreciate the time you and the reviewers spent on the revision of this study. We also would like to express our gratitude for the pertinent comments you have made on our work.

We look forward to hearing from you.

Yours sincerely,

Maria de Lourdes Teixeira de Moraes Polizeli, PhD

Lines 36, 38, 40, 41, 42, 44, 47, 49, 51, 54, 57-59, 60, 63-65, 70, 71, 82, 95, 107, 116, 127, 158, 184, 199, 206, 229, 232, 234, 238, 239, 248, 253-255, 259, 266, 277, 285, 296, 298, 300, 310, 312, 317-319, 321, 322, 328, 329, 356, 361: the number of the reference was added.

Lines 71-77: were rewritten;

Lines 98-102: were rewritten;

Lines 369-382: were rewritten;

Lines 442-444: the reference was added;

Lines 450-451: the reference was added;

Lines 458-460: the reference was added.

Response to reviewers’ comments:

Reviewer 01:

In this paper, carbohydrate-active enzymes (CAZymes) by Thermothelomyces thermophilus cultivated with Tamarindus indica seeds were investigated using the analysis of the secretome. Although some results seemed to be interesting, a few questions have to be clarified.

R.: The authors appreciate you for your time and review; and hope to promptly respond to the reviewer's suggestions and considerations.

  1. In Section 2.2, 0.1% yeast extract and 1% tamarind seeds were added into the medium. Because the addition of different nitrogen-containing substances in the medium can induce the production of different enzyme systems, and tamarind seeds also contain high protein content, these protein nutrients may affect the production of enzyme types. How to solve this problem?

R.: Yeast extract was used to produce the proteins necessary for the growth of the Thermothelomyces thermophilus, including structural proteins. Tamarind seeds, despite having nitrogen in their composition, were used especially as a carbon source, therefore, their main objective was to stimulate the production of CAZymes that degrade the cell wall found in this type of lignocellulosic biomass.

  1. As the title of the paper is for revealing “CAZymes for degradation of lignocellulosic biomass”, why not directly add tamarind carbohydrate to the chemical synthesis medium to avoid interference caused by the presence of protein in tamarind seeds?

R.: The choice to use tamarind seeds instead of the direct use of tamarind carbohydrate was due to two factors: 1) tamarind seeds are waste from the fruit pulp industry (Ramesh et al. 2015; Contato et al., 2023); 2) As they are considered agriculture residues, their use is inexpensive, unlike the use of commercially purchased tamarind carbohydrates.

Ramesh T, Rajalaksmi N, Dhatathrevan KS. 2015. Activated carbons derived from tamarind seeds for hydrogen storage. J. Energy Storage 4: 89-95. https://doi.org/10.1016/j.est.2015.09.005

Contato AG, Nogueira KMV, Buckeridge MS, Silva RN, Polizeli MLTM (2023) Trichoderma longibrachiatum and Thermothelomyces thermophilus co-culture: improvement the saccharification profile of different sugarcane bagasse varieties. Biotechnol Lett 45:1093-1102. https://doi.org/10.1007/s10529-023-03395-7

  1. In line 102-103, the Erlenmeyer flasks were incubated at 40 ℃ under static conditions for 72 h to analyze the composition of related enzymes, which was the best conditions for protein induction in the previous studies, and the data to prove this result should be reflected.

R.: The data to prove this result was best reflected in lines 98-102.

Lines 98-102: The Erlenmeyer flasks were incubated at 40 °C under static conditions for 72 h, the best conditions for protein induction described by Contato et al. (2021), which showed that these conditions were substantially better than in shaken cultivation and with shorter (24 and 48 h) or longer (96 h) times for evaluating enzyme production.

Reviewer 2 Report

Comments and Suggestions for Authors

The experimental article “Secretome analysis of Thermothelomyces thermophilus LMBC cultivated with Tamarindus indica seeds reveals CAZymes for degradation of lignocellulosic biomass” is devoted to a description of the secretome analysis, which allows us to determine the types of enzymes that are secreted by different microorganisms. The focus of these enzymes is determined to contribute to the breakdown of lignocellulosic raw materials. Enzymatic hydrolysis of lignocellulose continues to be a hot topic in fundamental biotechnology, so the article will be in demand among readers. According to all criteria (title, keywords, terminology and main points of the research), this article corresponds to the publication of J. Fungi. Despite the very rich experimental material on the classification of enzymes, the authors very compactly and clearly described the progress of the research and discussed the results obtained. In addition, the positive aspects of the article include the tabular structuring of the results obtained, allowing readers to trace the course of reasoning leading to the author’s conclusions, as well as a constant comparison of the results obtained with previously published ones. 59 references are involved, all exclusively on the substance of the issue, among them only three articles from the 2023 edition. A careful reading of the text reveals a number of comments that need to be addressed. The comments are listed.

Notes:

1. To enhance the relevance of the topic under discussion, include 2-3 reviews from 2023 in the introduction for substantive citation.

2. Articles for citation must be numbered according to the text.

3. Introduction, lines 79-80. It is necessary to provide a link confirming the expression “... using tamarind seeds as carbon source, which is a waste from the fruit pulp industry.”

4. Introduction, recommendation to formulate the purpose of the research in such a way that the conclusions demonstrate the achievement of this goal.

5. Lines 186-189. The article is missing Figure 1. The situation needs to be corrected.

6. It is necessary to expand the conclusions in the article and avoid overlap between the conclusions and the abstract.

Author Response

January 06th, 2024

Dear Mr. Demir Li,

Assistant Editor, Journal of Fungi,

Find attached the revised version of the manuscript entitled “Secretome analysis of Thermothelomyces thermophilus LMBC 162 cultivated with Tamarindus indica seeds reveals CAZymes for degradation of lignocellulosic biomass”. We appreciate the time you and the reviewers spent on the revision of this study. We also would like to express our gratitude for the pertinent comments you have made on our work.

We look forward to hearing from you.

Yours sincerely,

Maria de Lourdes Teixeira de Moraes Polizeli, PhD

Lines 36, 38, 40, 41, 42, 44, 47, 49, 51, 54, 57-59, 60, 63-65, 70, 71, 82, 95, 107, 116, 127, 158, 184, 199, 206, 229, 232, 234, 238, 239, 248, 253-255, 259, 266, 277, 285, 296, 298, 300, 310, 312, 317-319, 321, 322, 328, 329, 356, 361: the number of the reference was added.

Lines 71-77: were rewritten;

Lines 98-102: were rewritten;

Lines 369-382: were rewritten;

Lines 442-444: the reference was added;

Lines 450-451: the reference was added;

Lines 458-460: the reference was added.

Response to reviewers’ comments:

Reviewer 02:

The experimental article “Secretome analysis of Thermothelomyces thermophilus LMBC cultivated with Tamarindus indica seeds reveals CAZymes for degradation of lignocellulosic biomass” is devoted to a description of the secretome analysis, which allows us to determine the types of enzymes that are secreted by different microorganisms. The focus of these enzymes is determined to contribute to the breakdown of lignocellulosic raw materials. Enzymatic hydrolysis of lignocellulose continues to be a hot topic in fundamental biotechnology, so the article will be in demand among readers. According to all criteria (title, keywords, terminology and main points of the research), this article corresponds to the publication of J. Fungi. Despite the very rich experimental material on the classification of enzymes, the authors very compactly and clearly described the progress of the research and discussed the results obtained. In addition, the positive aspects of the article include the tabular structuring of the results obtained, allowing readers to trace the course of reasoning leading to the author’s conclusions, as well as a constant comparison of the results obtained with previously published ones. 59 references are involved, all exclusively on the substance of the issue, among them only three articles from the 2023 edition. A careful reading of the text reveals a number of comments that need to be addressed. The comments are listed.

R.: The authors appreciate you for your time and review; and hope to promptly respond to the reviewer's suggestions and considerations.

Notes:

  1. To enhance the relevance of the topic under discussion, include 2-3 reviews from 2023 in the introduction for substantive citation.

R.: Three (3) reviews from 2023 were added in the Introduction to enhance the relevance of the topic under discussion.

Added revisions:

  1. Wijayawardene NN, Boonyuen N, Ranaweera CB, de Zoysa HK, Padmathilake RE, Nifla F, Dai DQ, Liu X, Suwannarachi N, Kumla J, Bamunuarachchige TC, Chen, HH (2023) OMICS and other advanced technologies in mycological applications. J Fungi 9:688. https://doi.org/10.3390/jof9060688

  1. Salazar-Cerezo S, de Vries RP, Garrigues S (2023) Strategies for the development of industrial fungal producing strains. J Fungi 9:834. https://doi.org/10.3390/jof9080834

  1. Yang J, Yue HR, Pan LY, Feng JX, Zhao S, Suwannarangsee S, Champreda V, Liu CG, Zhao XQ (2023) Fungal strain improvement for efficient cellulase production and lignocellulosic biorefinery: Current status and future prospects. Bioresour Technol 385:129449. https://doi.org/10.1016/j.biortech.2023.129449

  1. Articles for citation must be numbered according to the text.

R.: The articles for citation were be numbered according to the text.

  1. Introduction, lines 79-80. It is necessary to provide a link confirming the expression “... using tamarind seeds as carbon source, which is a waste from the fruit pulp industry.”

R.: A reference confirming the expression was added.

Lines 71-75: In this context, this study aimed to report the elucidation of the secretome profile and categorization of CAZymes by function and family of the filamentous fungus T. thermophilus LMBC 162 cultivated by submerged fermentation using tamarind seeds as a carbon source, which is a residue from the fruit pulp industry [28].

  1. Introduction, recommendation to formulate the purpose of the research in such a way that the conclusions demonstrate the achievement of this goal.

R.: The purpose of the research was reformulated in the Introduction.

Lines 71-77: In this context, this study aimed to report the elucidation of the secretome profile and categorization of CAZymes by function and family of the filamentous fungus T. thermophilus LMBC 162 cultivated by submerged fermentation using tamarind seeds as a carbon source, which is a residue from the fruit pulp industry [28]. Obtaining these data, it was determined by relative abundance, which are the main proteins responsible for the degradation of the biomass bulk: cellulose, hemicellulose, and pectin.

  1. Lines 186-189. The article is missing Figure 1. The situation needs to be corrected.

R.: The Figure 1 was added.

  1. It is necessary to expand the conclusions in the article and avoid overlap between the conclusions and the abstract.

R.: The Conclusions in the article were expanded to avoid overlap with the Abstract.

Lines 369-382: The secretome analysis of T. thermophilus LMBC 162 cultivated by submerged fermentation with tamarind seeds, an abundant residue from the fruit pulp industry, reveals seventy-nine (79) CAZymes diversified into the five classes of CAZy database: 5.55% AAs; 1.48% CBMs; 2.58% CEs, 20.58% PLs; and 70.55% GHs, these being 54.97% cellulolytic GHs, 15.51% hemicellulolytic GHs, and 0.05 classified as other GHs. Between them, sixteen (16) CAZymes have CBMs, protein domains found in CAZymes, whose main role is to recognize and bind specifically to carbohydrates. In sum, CBMs are present in 48.74% of the proteins found in the T. thermophilus secretome profile, a value like that showed in the literature for other microorganisms. Observing the relative abundance, it is possible to state that only thirteen (13) proteins comprise 92.19% of the identified proteins secreted, and they are probably the main proteins responsible for the degradation of the bulk of the biomass: cellulose, hemicellulose, and pectin. The findings of this work allow to say that tamarind seeds are a residue option for the identification and production of lignocellulosic CAZymes.

Round 2

Reviewer 1 Report

Comments and Suggestions for Authors

None